# Effectiveness of Mental Health Literacy Programs in Primary and Secondary Schools: A Systematic Review with Meta-Analysis

**DOI:** 10.3390/children9040480

**Published:** 2022-03-31

**Authors:** Isaac Daniel Amado-Rodríguez, Rocio Casañas, Laia Mas-Expósito, Pere Castellví, Juan Francisco Roldan-Merino, Irma Casas, Lluís Lalucat-Jo, Mª Isabel Fernández-San Martín

**Affiliations:** 1PhD Program in Biomedical Research Methodology and Public Health, Autonomous University of Barcelona, 08193 Barcelona, Spain; 2Associació Centre d’Higiene Mental Les Corts, Research Department, 08029 Barcelona, Spain; rocio.casanas@chmcorts.com (R.C.); laia.mas@chmcorts.com (L.M.-E.); lluis.lalucat@chmcorts.com (L.L.-J.); 3Department of Medicine, Universitat Internacional de Catalunya (UIC), Sant Cugat del Vallès, 08195 Barcelona, Spain; pere.castellvi.obiols@gmail.com; 4Campus Docent, Sant Joan de Déu-Fundació Privada, Escuela de Enfermería, 08034 Barcelona, Spain; jroldan@santjoandedeu.edu.es; 5Program in Biomedical Research Methodology and Public Health, Autonomous University of Barcelona, 08193 Barcelona, Spain; irma.casas@uab.cat; 6Unitat Docent Multiprofessional Gerència Territorial Barcelona, Institut Català de la Salut, 08025 Barcelona, Spain; mifsanmartin.bcn.ics@gencat.cat

**Keywords:** mental health literacy, stigma, help-seeking, adolescent, intervention, meta-analysis, systematic review

## Abstract

In recent years, there has been an increase in studies evaluating the effectiveness of mental health literacy programs within the context of education as a universal, preventive intervention. A systematic review and meta-analysis regarding the effectiveness of mental health literacy interventions in schools, from 2013 to the present, on mental health knowledge, stigma, and help-seeking is conducted. Of the 795 identified references, 15 studies met the inclusion criteria. Mental health knowledge increased after the interventions (standardized mean difference: SMD = 0.61; 95% CI (0.05, 0.74)), at two months (SMD = 0.60; 95% CI (0.4, 1.07)) and six months (SMD = 0.39; 95% CI (0.27, 0.51)). No significant differences were observed between stigma and improving help-seeking. Mental health literacy interventions are effective in augmenting mental health knowledge, but not in reducing stigma or improving help-seeking behavior.

## 1. Introduction

Most mental health problems in individuals commence between 12 and 25 years of age, with 20% of adolescents being affected [1,2]. Moreover, it is calculated that 50% of such issues appear before the age of 14 [3]. In spite of this elevated incidence, 70–80% of young people and adults do not receive adequate mental healthcare [4]. Due to the importance of mental health during adolescence, international bodies such as the European Union (EU) and the World Health Organization (WHO) are promoting policies and interventions to prevent mental health disorders and raise awareness. They are highlighting the need to analyze and diffuse good practice, for instance, with the EU-Compass for action on Mental health and Well-being [5], and encourage multi-sectorial strategies to provide integral support for mental health [6]. The WHO is currently carrying out its Comprehensive Mental Health Action Plan 2013–2020 and its subsequent update until 2030. This plan aims to ensure that 80% of countries will have at least three functioning national, multisectoral mental health promotion and prevention programs by 2030. These plans have to be targeted at vulnerable groups using, as an example, programs like mental health awareness/anti-stigmatization or school-based mental health prevention and promotion [7,8]. Within this framework lies the concept of mental health literacy (MHL), which has been described by Jorm [9] as “the knowledge and beliefs about mental health problems that help in their recognition, management, and prevention”. This definition includes the capacity to recognize and inform oneself about mental disorders and their risk factors and how to seek treatment and professional support [9]. In recent years, the notion of MHL has been undergoing changes and has become an integral, multifactorial concept that recognizes the need to reduce mental health stigma and foster first aid [10,11,12]. Its programs are focused on the capacity of individuals to understand and make correct use of healthcare information so as to better understand mental health disorders, improve treatment compliance, and be aware of how and when to look for assistance. The effectiveness of MHL interventions has been demonstrated in systematic revisions such as those by Wei [2] and Seedaket [13]. The former included 27 studies carried out up to 2012 in the field of education with 17,643 individuals aged 12 to 25 years. It reported an increase in knowledge concerning mental health and a decrease in stigma both in the short-term and during follow-up. The latter was made up of seven randomized clinical trials (RCTs) performed up to 2020 in a population aged 10 to 19 years and had results that were similar to Wei’s review. All the interventions improved mental health knowledge: 71% and 25% for stigma and help-seeking, respectively. As neither of the two systematic reviews included a meta-analysis, the effect size of this kind of intervention could not be established. The objective of this systematic review with meta-analysis is to evaluate the effectiveness of MHL interventions on mental health knowledge, stigma, and help-seeking in young, school-enrolled populations from the beginning of the international Comprehensive Mental Health Action Plan 2013–2030 to the present.

## 2. Materials and Methods

This review was carried out according to the protocol Preferred Reporting Items for Systematic reviews and Meta-Analyses (PRISMA) [14]. The review protocol was pre-registered on the International Prospective Register of Systematic Reviews (PROSPERO) [15] prior to data extraction (CRD42021281178, access date on 10 October 2021 in https://www.crd.york.ac.uk/prospero/display_record.php?RecordID=281178).

### 2.1. Inclusion Criteria

Inclusion criteria for studies regarding the efficacy/effectiveness of MHL interventions were the following: (1) population aged between 10–19 years; (2) research carried out in primary and/or secondary schools; (3) interventions with presence-based modality (school setting); (4) RCT and quasi-experimental (QE) design; (5) outcomes: mental health knowledge, stigma, and/or help-seeking; (6) studies performed from commencement of the WHO 2013–2030 Action Plan on Mental Health [7,8] and the set-up of programs of mental health prevention and promotion such as MHL; and (7) no language limits. Exclusion criteria were: (1) qualitative studies; (2) determined populations, for instance, university students and students with specific mental health or drug abuse issues; (3) studies concerning mental health first aid; (4) MHL interventions focused on a specific disorder (depression, schizophrenia...); and (5) MHL interventions that only assess the concept of stigma.

### 2.2. Study Search and Selection Strategy

The search was performed with the databases Pubmed, Psycinfo, Web of Science, and Cinahl from January 2013 to July 2020. The bibliographic search in the three databases included the following terms in the widest sense possible: (“mental health literacy” AND (“program*” OR “intervention*”) AND “young”) OR (“mental health literacy” AND school AND adolesc*). The secondary strategy involved examining the references of the included studies and relevant reviews. Duplicates and studies that did not fulfill the inclusion criteria were eliminated, and only article-type publications were selected. After revising the study abstracts, those that were related to the theme of the present research were accepted. Finally, a hand search was conducted. The studies were included according to peer review, and the following information extracted: author, year, country, design, educational stage, age, number of subjects, follow-up, type of intervention, duration of intervention, professional who delivers intervention, measurement tools, and level of evidence.

The Scottish Intercollegiate Guidelines Network (SIGN) [16] was employed to establish the identified studies’ level of scientific evidence. We completed a cross-tabulation of each article selected by I.A.R. response categories to obtain frequencies and percentages of each variable of interest in the study. Inter-rater agreement was performed among two team members (blinded to peer review) to assess acceptance by determining the grade of evidence and practice guideline levels. The two members extracted the data from the included investigations and coded the studies considering the variables of our research questions. Concordance was considered adequate if it met at least 80% [17]. Seven studies (46, 6%) were randomly selected, and the investigators independently extracted the information and entered it into a codebook. The consistency of these data among the two evaluators was evaluated by calculating the proportion of agreements and disagreements in the different codes. Interrater agreement was 97.4%. Cohen’s Kappa was not calculated due to the low variability of agreement between the evaluators [18]. The SIGN model is based on three sections (internal validity, general evaluation, and study description) and classifies the research into 8 levels of evidence. These range from 1++ the maximum available evidence (high-quality meta-analyses, systematic reviews of clinical trials, and clinical trials with a reduced risk of bias) to 4 (expert opinions) [16] (Appendix A, Table A1).

### 2.3. Analysis

A meta-analysis of the main variables (mental health knowledge, stigma, and help-seeking) was performed employing standard mean differences (SMDs) due to the different quantitative scales employed. The pooled estimates were evaluated post-intervention and at 2- and 6-months follow-up. A meta-analysis of random effects was carried out in order to obtain pooled estimates as differences were observed among the studies. These included: (a) content and methodology; (b) intervention duration; (c) outcome measures and definition criteria; (d) follow-up evaluation; and (e) clinical and sociodemographic characteristics. Opposite direction scale values were recodified.

Heterogeneity was measured by the statistical test I^2^. A value of 0–40% indicated low heterogeneity; 40–75% moderate; 75–100% considerable [19]. Publication bias was assessed with a funnel plot (scatter plot with treatment effectiveness against study sample size) [20]. Review Manager (RevMan) software was employed for the meta-analysis, and the program Stata was employed for the meta-regression.

## 3. Results

### 3.1. Study Description

Out of 795 identified studies, 15 fulfilled the inclusion criteria (Figure 1), 6 were RCT, and 9 QE. With respect to the country of origin, three were performed in Japan, three in Canada, two in the United Kingdom, two in the United States, and only one each for Spain, Portugal, Nigeria, Norway, and Australia. Of the 15 studies, 5 (33%) assessed the intervention’s effectiveness pre- and post-intervention and 10 (66.7%) provided follow-up evaluations ranging from 1 to 12 months (Table 1) [21,22,23,24,25,26,27,28,29,30,31,32,33,34,35].

Approximately half of the selected study interventions were carried out by teaching professionals in the educational centers (*n* = 8; 53.3%). Some disparity was observed regarding the length of the interventions, which varied from 45 min to 12 h. One of them did not specify its duration [27].

### 3.2. Study Quality

Table 1 depicts the level of scientific evidence of the identified studies. It ranged from 1++ (maximum level of evidence) to 2+ which were studies with a moderate probability of establishing causal relationships (1++: 20%, 1+: 13.33%, 1−: 6.67%, 2++: 33.33% and 2+: 26.67%). All the RCTs had a level of evidence between 1++ and 1−, whilst the QEs were classified between 2++ and 2+.

### 3.3. Evaluation of Mental Health Knowledge

Of the 15 selected studies, 1 of them pending publication [26], 6 presented statistically significant improvements at the moment of post-intervention and 10 at follow-up (between 1 and 12 months). In only one study was a statistically significant improvement observed at follow-up, but not at the moment of post-intervention [27].

With respect to the analysis of change in knowledge, we included eight studies (six RCTs and two QEs) at the moment of post-intervention and obtained, broadly speaking, a significant increase (SMD = 0.61; 95% CI (0.17, 1.04)) with marked heterogeneity I^2^ = 97% (Figure 2). If we analyze it according to the type of study, a significant increase was observed in both QEs (SMD = 1.26; 95% CI (0.59, 1.93)) and in RCTs (SMD = 0.40; 95% CI (0.05, 0.74)). At the two-month follow-up, we included three studies (one RCT and two QEs) and observed a significant rise in knowledge (SMD = 0.60; 95% CI (0.14, 1.07); I^2^ = 89%) (Figure 3). If we analyzed it according to the type of study, a significant increase was observed in the QEs (SMD = 0.85; 95% CI (0.72, 0.99)) but not in the RCTs (SMD = 0.08; 95% CI (−0.24, 0.41) *p* = 0.61). At the six-month follow-up, three RTCs were incorporated, which also showed a significant increase in knowledge (SMD = 0.39; 95% CI (0.27, 0.51)) without heterogeneity I^2^ = 0% (Figure 4). The QEs studies presented a greater SMD than that observed for the RCTs at both post-intervention and the two-month follow-up. After performing the meta-regression, we did not observe a significant relationship between the hours of intervention and the effect on knowledge (*p* = 0.450) (Table 2).

### 3.4. Evaluation of Mental Health Stigma/Attitudes

Of the selected studies, 12 evaluated changes in mental health stigma/attitudes, 9 of which reported statistically significant results [21,22,23,24,27,29,30,32,34]. With respect to the analysis of stigma, at both the moment of post-intervention (SMD = 0.06; 95% CI (−0.28, 0.41); I^2^ = 95%) and at 6 months (SMD = 0.012; 95% CI (−0.44, 0.68); I^2^ = 95%), no statistically significant differences were observed after the interventions (Figure 5 and Figure 6). We did not observe a significant relationship between the hours of intervention and the effect on stigma (*p* = 0.890) at the meta-regression (Table 3).

### 3.5. Evaluation of Help-Seeking

Assessment of help-seeking for mental health was present in nine studies, five of which reported a significant improvement at the moment of post-intervention [24,27,31,33,35].

With respect to the analysis of change in help-seeking, we included three studies (three RCTs) at the moment of post-intervention and did not obtained a significant increase (*p* = 0.24; SMD = 0.20; 95% CI (−0.13, 0.52)) with heterogeneity I^2^ = 90% (Figure 7). At the six-month follow-up we included two studies (two RCTs) and not statistically significant differences were observed after the interventions (SMD = 0.02; 95% CI (−0.14, 0.19) I^2^ = 0%) (Figure 8). After performing the meta-regression, we did not observe a significant relationship between the hours of intervention and the effect on help-seeking (*p* = 0.392) (Table 4).

The funnel plots were calculated to assess publication bias (Appendix B, Figure A1, Figure A2, Figure A3, Figure A4, Figure A5, Figure A6 and Figure A7), although statistical tests were not applied due to having less than 10 studies.

## 4. Discussion

This review examines the effectiveness of mental health literacy interventions for young people in educational centers from the beginning of the international Comprehensive Mental Health Action Plan 2013–2030 to the present. It assesses how such programs influence knowledge regarding mental health, the stigma surrounding it, and help-seeking. Most of them displayed positive effects, although only the variable of knowledge showed significant improvement in the short- and long-term.

With respect to knowledge, we observed that after analysis, the studies with the best post-intervention results were two QEs (Yamaguchi and Bella-Awusah) [28,35], followed by one RCT from Campos [24]. These findings were maintained at two months in Yamaguchi and Bella-Awusah [28,35] and at six months in Campos [24]. As shown in the comparison of results, QE studies tended to overestimate intervention effectiveness (lack of randomization leading to a greater probability of bias). The three interventions lasted less than 3 h and differed with respect to the characteristics of the individual responsible for carrying them out (teacher, researcher, and expert psychologist). In general, we observed a statistically significant improvement in knowledge for the studies analyzed both post-intervention (SMD = 0.61) and at the six-month follow-up (SMD = 0.39) compared to the usual programs.

Regarding stigma, the studies by Campos [24] and Perry [21] were the only ones that demonstrated statistically significant improvements post-intervention at the meta-analysis. While both were RCTs, they differed with respect to duration (3 h and 10 h, respectively) and professionals delivering the intervention (expert psychologists and teachers, respectively). Both studies did not present contact with a person who has suffered from mental disorders in comparison with the studies by Casañas [26], Pinto-Foltz [25], and Chisholm [23], who obtained worse results in this variable. Milin´s study [22] did not present direct contact, but it did have video interviews and cases with young people with mental illnesses. The QE study of Bella-Awusah [25] analyzed intervention carried out by a psychologist and without direct contact but did not present post-intervention improvements. Future research should assess whether a first-person experience is an unfavorable factor in reducing stigma or if educational sessions with specialists or teachers are more effective in reducing stigma. We observed that Campos [24] employed the same tool to measure knowledge and stigma in contrast to the other studies, which used different scales, this could be a bias in this study, and we should take these results with caution. After analyzing the studies, we did not find a significant improvement post-intervention (SMD = 0.06; *p* = 0.72) nor at the six-month follow-up (SMD = 0.12; *p* = 0.68).

In the help-seeking variable, the three studies included in the meta-analysis were RCTs, with a different duration between them (10 h [22], 4 h 45 min [23], and 3 h [24]), as those in charge of carrying out the interventions (teachers [22], a researcher together with teachers [23], and an expert psychologist [24]). We observed that Campos [24] presents a significant improvement in the short term in the meta-analysis but not at 6 months. In general, no significant improvements were found in this variable, neither post-intervention (SMD = 0.2; *p* = 0.24), nor at 6 months (SMD = 0.02; *p* = 0.79).

Regarding the analysis over time, worse results were observed in the three main variables as time progressed. This may be due to the fact that over time if knowledge is not reinforced, it is progressively forgotten, as Ebbinghaus [36] and other current studies postulate [37]. In addition, no significant time-dependent relationship was observed in the hours of intervention with the three main variables (knowledge, stigma, and help-seeking), as can be seen in the meta-regression in Table 2, Table 3 and Table 4.

These results are similar to the results of the Wei and Seedaket [2,13] reviews both in terms of knowledge, stigma, and help-seeking at a qualitative level. However, because they did not perform their respective meta-analyses, we cannot compare them. There was considerable variability amongst the interventions with respect to format, content, and measurement tools which hindered comparison. Moreover, their duration ranged from 45–50 min, as was the case of the Short MHL program [31,33], up to 12 h for The Guide [29,30]. In addition, MHL domains were not consistent; some studies included them in their broadest sense, for instance, Perry [21], Campos [24], and Pinto-Foltz [23]; whilst others were only concerned with the detection of one or two specific mental health disorders such as Ojio [31,33].

Our findings concur with the systematic review performed by Mansfield [38], in which the lack of homogeneity regarding the concept of MHL was highlighted. There was also a tendency to focus solely on the identification of mental health disorders rather than examining the issue from a broader, more positive perspective. The review of Nobre [39] provided us with plausibility to the search carried out since they present similar results in the selected articles, despite having different objectives and selection criteria. The most recent review, Freitan [40], also assesses knowledge and stigma but not help-seeking. Freitan’s analysis differs from ours by including participants with mental illness, the time stage of the included studies, and the interventions outside the school context. Both studies present after the meta-analysis improvement in knowledge but not in stigma in the follow-up phase.

Our review presents some limitations. Firstly, the design of the studies was heterogeneous, and we observed a greater weighting in the case of the QEs when calculating the intervention effect. Moreover, the three variables (knowledge concerning mental health, stigma, and help-seeking behavior) were assessed with different scales, and one-third of the studies employed their own unvalidated tools to measure help-seeking. Secondly, regarding publication bias, it should be noted that the statistical analysis could not be performed due to having fewer than 10 studies. We have to emphasize that only one study [28] belongs to an economically depressed region. Instead, we found descriptive studies in those locations like Vietnam, Sub-Saharan Africa, and Zambia [41,42,43], but not the evaluation of interventions. This can cause the results to be irreproducible in these regions. Thirdly, it should be noted as a limitation that studies prior to 2013 were not included. In this study, we were interested in assessing the studies carried out during the period of implementation of the WHO Comprehensive Mental Health Action Plan 2013–2020 [7] and the promotion of this plan in mental health prevention and promotion interventions such as MHL in the adolescent population. We have to add that our data search was mainly in health databases. This could have caused the loss of information from specialized education databases.

Our results should be interpreted with caution. In spite of the short- and medium-term clinical benefits regarding the improvement in knowledge, we lack sufficient data to evaluate the efficacy of these interventions long-term (12 months follow-up). Further studies with more robust methodology, for instance, RCTs, with interventions that share the same MHL concept domains and employ the same measurement tools (particularly with respect to help-seeking) are warranted. More studies are needed that focus on the conditions in which the study is carried out, such as the content of the intervention, the person responsible for delivering the intervention, or the way of carrying out the intervention (use of videos, contact with first-person experience, talking, etc.). Subsequent studies should assess the following time frame until 2030 to study the evolution of the effectiveness of interventions based on MHL in the classroom after the end of the WHO plan in that year.

## 5. Conclusions

Mental health literacy interventions in a young school-age population improve mental health knowledge. Considering the limited data meta-analyzed, we cannot conclude that MHL interventions are effective on stigma and help-seeking outcomes. Even so, the results of individual studies on such outcomes are encouraging. There is considerable variability in the concept of mental health literacy and the instruments used to evaluate these interventions. Future research should go further on the effectiveness of MHL intervention considering stigma and help-seeking outcomes and, in addition, analyze other factors associated with the effectiveness of MHL interventions, such as the length of follow-ups and the type of professional in charge of carrying it out.

## Figures and Tables

**Figure 1 children-09-00480-f001:**
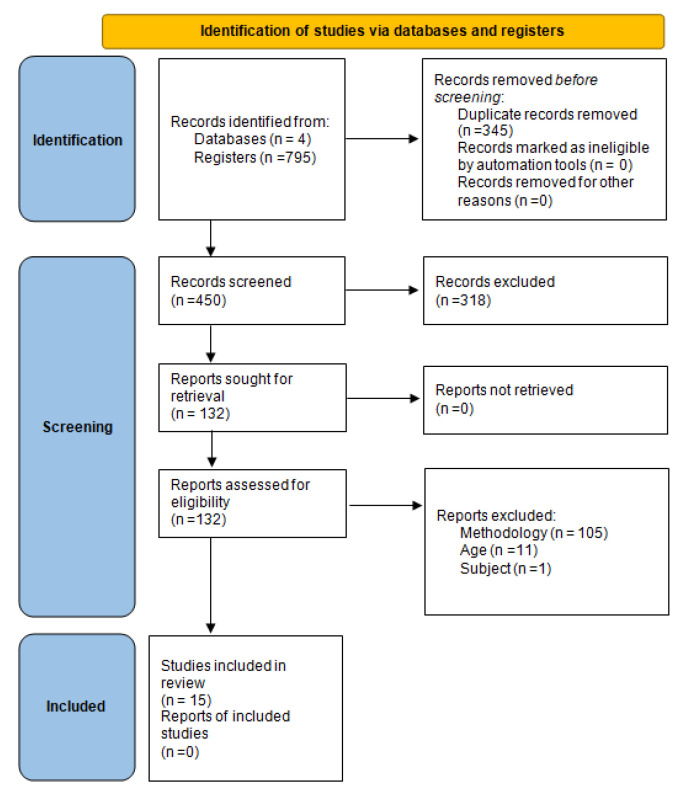
PRISMA 2020 flow diagram for new systematic reviews, which included searches of databases and registers.

**Figure 2 children-09-00480-f002:**
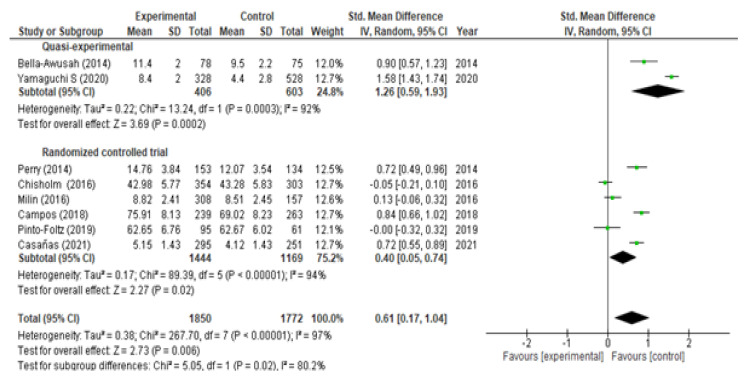
Meta-analysis of change in level of knowledge post-intervention.

**Figure 3 children-09-00480-f003:**
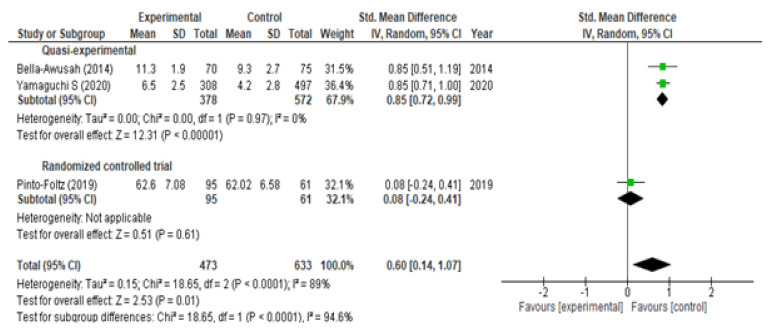
Meta-analysis of change in level of knowledge at 2 months.

**Figure 4 children-09-00480-f004:**
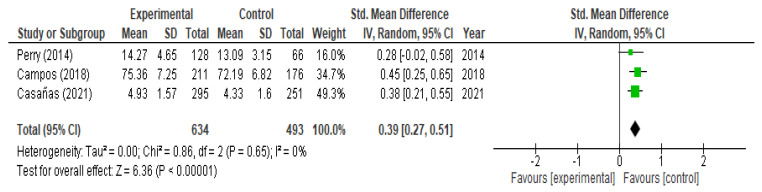
Meta-analysis of change in level of knowledge at 6 months.

**Figure 5 children-09-00480-f005:**
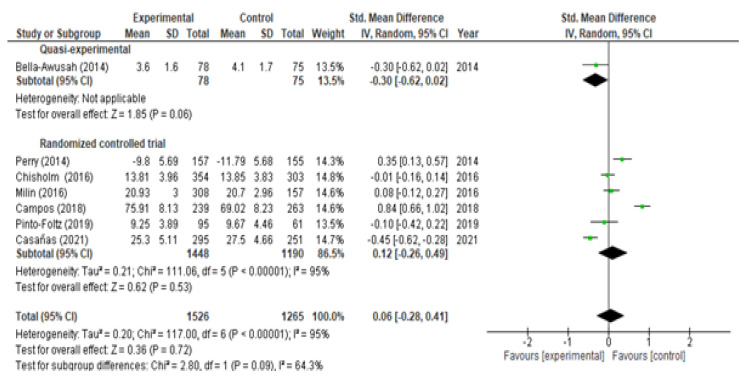
Meta-analysis of change in level of stigma post-intervention.

**Figure 6 children-09-00480-f006:**
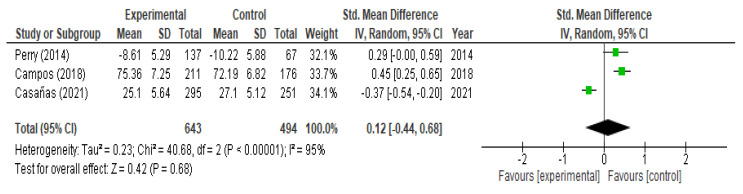
Meta-analysis of change in level of stigma at 6 months.

**Figure 7 children-09-00480-f007:**
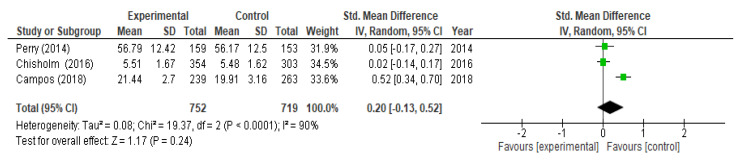
Meta-analysis of change in help-seeking post-intervention.

**Figure 8 children-09-00480-f008:**
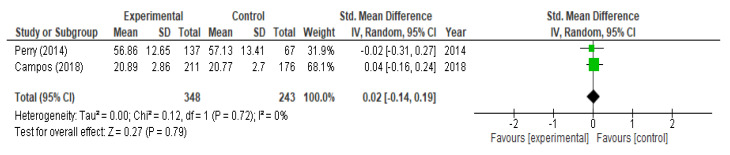
Meta-analysis of change in help-seeking at 6 months.

**Table 1 children-09-00480-t001:** Description of the studies included in the systematic review.

Author (Year) Country	Design	Education Stage	Age	N	Follow-Up	Intervention	Evaluation Tools	Sign
Perry (2014) [21] Australia	RCT	Secondary	13–16 years	IG (*n* = 207) CG (*n* = 173)	Pre, post, and at 6 months	IG: Headstrong (10 h): Given by teachers at the educational center CG: Usual intervention	Depression Literacy Scale (D-Lit) Depression Stigma Scale (DSS) Inventory of Attitudes towards Seeking Mental Health Services (IASMHS)	1++
Milin (2016) [22] Canada	RCT	Secondary	16–18 years	IG (*n* = 362) CG (*n* = 172)	Pre and post (between 1 and 2 months)	IG: The Guide (6 h): Given by teachers at the educational center CG: Usual intervention	Questionnaire about mental health knowledge: 15 multiple-choice questions (OC) Stigma: Likert scale with 8 questions (OC)	1+
Chisholm (2016) [23] United Kingdom	RCT	Secondary	12–13 years	IG (*n* = 354) CG (*n* = 303)	Pre and post at 2 weeks	IG: School Space (4 h and 45 mins): Given by researchers and teachers at the educational center in contact with a patient; CG: Change contact for mental health record	Mental Health Knowledge Schedule (MAKS) The Reported and Intended Behaviour Scale (RIBS) Seeking Help (OC)	1++
Campos (2018) [24] Portugal	RCT	Secondary	12–14 years	IG (*n* =259) CG (*n* =284)	Pre, post at 1 week, and 6 months	IG: Finding Space (3 h): Given by expert psychologist and student of Psychology Master’s Degree; CG: Usual intervention	Mental Health Literacy questionnaire: 18 items about knowledge/stereotypes, 10 items about seeking help, and 5 items about self-help strategies	1+
Pinto-Foltz (2019) [25] United States	RCT	Secondary	13–17 years	IG (*n* = 95) CG (*n* = 61)	Pre, post, at 1 and 2 months	IG: In Our Own Voice (1 h): Given by individuals with mental health problems (older than 18 years); CG: No intervention	In Our Own Voice Knowledge Measure The Revised Attribution Questionnaire	1−
Casañas (2021) [26] Spain	RCT	Secondary	13–15 years	IG (*n* = 295) CG (*n* = 251)	Pre, post, at 6 and 12 months	IG: EspaiJove.net Program (7 h): Given by mental health nurses CG: No intervention	Questionnaire about knowledge EspaiJove.net: 35 multiple-choice items The Reported and Intended Behaviour Scale (RIBS) General Help-Seeking behaviors modified	1++
Skre (2013) [27] Norway	QE	Secondary	13–16 years	IG (*n* = 520) CG (*n* = 550)	Pre and post at 2 months	IG: Mental health for everyone (3 days): Given by teachers at the educational center; CG: Usual intervention	Knowledge about mental disorders scale Prejudice and stigma: Likert scale of 4 items (OC) Seeking help: free response questions (OC)	2++
Bella-Awusah (2014) [28] Nigeria	QE	Secondary	10–18 years	IG (*n* = 78) CG (*n* = 76)	Pre, post, and at 6 months	Dogra 2005 Adaptation Program (3 h): Given by researchers CG: Usual intervention	UK Pinfold questionnaire modified (OC)	2+
Mcluckie (2014) [29] Canada	QE without CG	Secondary	14–15 years	*n* = 265	Pre, post, and at 2 months	The Guide (10–12 h): Given by teachers at the educational center	Questionnaire about knowledge: 28 items (OC) Attitudes/stigma: Likert scale 8 questions (OC)	2++
Kutcher (2015) [30] Canada	QE without CG	Secondary	14–15 years	*n* = 175	Pre, post, and at 2 months	The Guide (10–12 h): Given by teachers at the educational center	Questionnaire about knowledge: 28 items (OC) Attitudes/stigma: Likert scale 8 questions (OC)	2+
Ojio(2015) [31] Japan	QE without CG	Secondary	14–15 years	*n* = 118	Pre, post, and at 3 months	The Short MHL Program for Teens (1 h and 40 min): Given by teachers at the educational center	Questionnaire about knowledge: 12 questions and 2 comic strips with 3 questions about seeking help (OC)	2++
Patalay (2017) [32] United Kingdom	QE without CG	Secondary	13–15 years	*n* = 234	Pre and post	Open Minds (1 h and 40 min): Given by medical students	Questionnaire about knowledge: 4 questions (OC) Non-stigmatized attitudes: 5 items (OC)	2++
Ojio (2019) [33] Japan	QE without CG	Primary	10–12 years	*n* = 662	Pre, post, and at 3 months	The Short MHL Program for Pre-Teens (45 min): Given by teachers at the educational center	Questionnaire about knowledge: 7 questions (OC) A comic strip asking what they would do if it happened to them (OC)	2+
Lindow (2020) [34] United States	QE without CG	Secondary	12–18 years	*n* = 436	Pre and post at 3 months	Youth Aware of Mental Health (4 h and 10 min: Given by personnel trained by researchers at the educational center	General help-seeking behaviors questionnaire (GHSQ) Stigma: 4 and 7 items (OC) Mental health resources: 2 scales (OC)	2+
Yamaguchi (2020) [35] Japan	QE	Secondary	15–16 years	IG (*n* = 364) CG (*n* = 611)	Pre, post, and at 2 months	IG: Short MHL Program (SMHLP) (50 min): Given by teachers at the educational center CG: Usual intervention	Questionnaire about knowledge: 10 questions (OC) Recognition and intention to seek help: 2 comic strips ad 2 questions with 4 levels (OC)	2++

Abbreviations: RCT, randomized clinical trial; QE, quasi-experimental; CG, control group; IG, intervention group; *n*, number of participants; OC, own creation.

**Table 2 children-09-00480-t002:** Meta-regression between hours of intervention and knowledge.

StMeanDiff	Coef.	Std. Err.	t	P > |t|	[95% Conf. Interval]
Length	−0.031	0.040	−0.78	0.450	−0.118; 0.056
Cons	0.691	0.213	3.24	0.007	0.226; 1.155
REML estimate of between-study variance			Number of obs = 14		
% residual variation due to heterogeneity			Tau2 = 0.2005		
Proportion of between-study variance explained			I-squared res = 95.15%		
With Knapp-Hartung modification			Adj R-squared = −2.42%		

**Table 3 children-09-00480-t003:** Meta-regression between hours of intervention and stigma.

StMeanDiff	Coef.	Std. Err.	t	P > |t|	[95% Conf. Interval]
Length	−0.007	0.048	−0.14	0.890	−0.118; 0.104
Cons	0.118	0.298	0.39	0.704	−0.571; 0.806
REML estimate of between-study variance			Number of obs = 10		
% residual variation due to heterogeneity			Tau2 = 0.1778		
Proportion of between-study variance explained			I-squared res = 94.43%		
With Knapp-Hartung modification			Adj R-squared = −12.22%		

**Table 4 children-09-00480-t004:** Meta-regression between hours of intervention and help-seeking.

StMeanDiff	Coef.	Std. Err.	t	P > |t|	[95% Conf. Interval]
Length	−0.033	0.033	−1.00	0.392	−0.138; 0.072
Cons	0.325	0.221	1.46	0.239	−0.381; 1.03
REML estimate of between-study variance			Number of obs = 5		
% residual variation due to heterogeneity			Tau2 = 0.0426		
Proportion of between-study variance explained			I-squared res =82.93%		
With Knapp-Hartung modification			Adj R-squared = 0.61%		

## Data Availability

The data presented in this study are available on request from the corresponding author.

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
