# Peer review of "Effectiveness of Mental Health Literacy Programs in Primary and Secondary Schools: A Systematic Review with Meta-Analysis"

_children, 2022, doi:10.3390/children9040480_

Round 1
Reviewer 1 Report
I thank the Authors for their corrections, additions and responses. I have no further comments.
Author Response
We appreciate very much this comment. Thanks for the effort and the evaluation

Reviewer 2 Report
I want to thank the authors for allowing me to review their work. The authors have done an excellent job in the introduction and have provided enough information about why they felt like doing this vital study.
Methods: The authors did a systematic review with meta-analysis
I like the discussion part, but I suggest that the authors modify their conclusion part. They say mental health literacy in a young, school-aged population improves mental health knowledge but not stigma or help-seeking behaviors. So I would suggest the authors modify the statement and say something like with the limited data they analyzed, they are not able to conclude that mental health literacy improves stigma or help-seeking behaviors as we know with MHL we have seen an improvement in stigma and help-seeking behaviors in this group of individuals.
Author Response
We appreciate very much this comment. We have added the following changes to the conclusions section:
"Mental health literacy interventions in a young school-age population improve mental health knowledge. Considering the limited data meta-analysed, we cannot conclude that MHL interventions are effective on stigma and help-seeking outcomes. Even so, the results of individual studies on such outcomes are encouraging. There is considerable variability in the concept of mental health literacy and the instruments used to evaluate these interventions. Future research should go further on the effectiveness of MHL intervention considering stigma and help-seeking outcomes and, in addition, analyze other factors associated with the effectiveness of MHL interventions such as the length of follow-ups and the type of professional in charge of carrying it out"

This manuscript is a resubmission of an earlier submission. The following is a list of the peer review reports and author responses from that submission.
Round 1
Reviewer 1 Report
This manuscript reports the results of a well-conducted and well-written systematic review and meta-analysis on the effectiveness of mental health literacy interventions for children and adolescents in school settings. Results regarding acquired knowledge, stigma and help seeking, post-intervention and months later are meta-analyzed. I recommend the publication of this work, after having clarified the following points:
- It is recommended to publish the review and meta-analysis protocol on platforms such as Prospero, before starting the screening or in any case before the final analyses. Was this done for this study? If not, I recommend adding it to the limitations, and if so please write the registration number in the methods.
- The inclusion criteria report two reasons for the inclusion of studies only starting from 2013. The first reason (date of the last review available) seems out of place: just because a review was already available does not mean that the studies included in it they had to be excluded (indeed, they would constitute added value). I recommend removing this motivation. On the other hand, the motivation of the WHO program seems to me much more relevant. I would emphasize this in the introduction, devoting a paragraph to the authors' considerations of why studies released under this WHO program should be studied and considered separately.
- Exclusion criteria n.3 is redundant and already specified in the inclusion criteria, it should be removed.
- “databases” is a single word, please correct occurrences of “data bases”.
- A great limitation of this study is represented by the absence of moderation analysis (meta-regression). Factors such as follow-up duration and length of the intervention could be included as predictors in the meta-analysis, in order to adjust all effects and obtain more solid results without the need for discretization. Are the Authors able to provide these additional analyses. If not, such absence should be noted in the limitations.
- I am confused about the following statement: “ It has not been possible to carry out a meta-analysis of the variable help-seeking as different instruments were employed”: was this not true for other measures, such as knowledge? Looking at table 1, it seems that even other outcomes such as knowledge were reported using different and ad-hoc instruments, however the effects were meta-analysed using appropriate effect measures (SMD). Please clarify this point.
- The study by Casanas is reported as both Casanas 2020 and Casanas 2021, please correct the inconsistency.
- The study by Casanas 2020/2021 is not properly referenced: the wrong citation is used (2018, protocol only). Please correct this. Interestingly, it seems that those Authors found a negative effect of the intervention on stigma, how was that commented and discussed? This could be added in the discussion, as it could explain the negative finding about stigma.
Reviewer 2 Report
Thank you for inviting me to review this interesting meta-analysis of mental health literacy interventions. This is an important and growing area, and so this summary is potentially useful. There are some excellent facets of this work - the introduction clearly describes how the concept of mental health literacy has changed over the years and how the authors planned to define it. The review involves a quality assessment and provides a PRISMA diagram. The authors clearly define how they will apply the I2 statistic (written as I2 throughout but this may be housestyle) - the paper could be improved by consideration of the points below
- The abstract and conclusion state that these interventions were ineffective for improving help seeking when in fact the data were impossible to meta-analyse (but could they be described narratively instead ?)- this should be softened to no evidence that they improve helpseeking
- The reference for how many young people access mental health care is a secondary source and is misquoted - 70-80% of young people DO NOT access mental health care despite documented needs
- While the introduction very clearly describes the definition of mental health literacy that will be applied - it fails to explicitly set up the current study and how it fits with the two reviews mentioned in the introduction - the current paper is clearly aimed to update one of the authors earlier work - and implicitly the second review mentioned only includes trials - but this should be made more explicit and clearer - I also wondered why the papers retrieved in the current study were not added to those from the earlier SR to improve the metaanalysis?
- Given that the interventions were delivered in the education setting, I wondered why only health databases were searched - might the authors have missed some papers published in these journals? Also was there any effort to search trial registries or contact experts in the field to ensure that all papers were retrieved? There was backward citation and hand searching (though the authors need to report which journals were hand searched) but what about forward citation chasing? the discusison should include some discussion on the chances of missing publications given how many were detected by supplementary searches (which should also be included in the PRISMA flow diagram) - the use of automatic tools to remove ineligible mentioned on this diagram should be reported explicitly in the text
- I wondered why there was no metaregression by potentially important factor - education stage obviously could not be conducted, but length of intervention, length of follow up (might allow all follow ups after immediate post intervention to be combined rather than splitting into 3 to 6 months) and who delivered if sufficient variability - if insufficient data the discussion should discuss what might be the important variables to discuss
- The quality of studies does not seem to be poor - but this is not a scale I am familiar with so qualifying the ranges reported with the maximum and minimum possible would be helpful - also if there were particular methodological issues that others could avoid that would be worth a comment in the results and picking up in the discussion
- Where there any exclusions based on setting (all are Higher Income Countries - were there no studies in LMICs or was this an exclusion) or language (English language only?)
-
The method of - I think- title/abstract and then full text screening - as described in the PRISM flow diagram needs clarifying in the text - how many titles and abstracts were double screened for reliability and were any full texts and data extraction pro forma also double screened - how were disagreements resolved and what was the level of agreement between raters - clarity is obscured by some odd choices of words
ie page 2 penultimate sentence of the procedure - revising? do the authors mean revising the inclusion criteria if so how and why - or do they mean reviewing the results of two independent screens? Same paragraph - peer review - do the authors mean consensus by discussion amongst the researchers to resolve disagreement
- was the review registered on PROSPERO? and is the protocol and search strategy available publically if not?
- Table 1 is excellent but it would be good to include data on the length of the intervention and who delivered this as this is mentioned in the discussion without supporting information - details about what was covered in the manual / curriculum (or that this is not available) would also have been informative
- First sentence of page 6 - 53% does not equal "most" - approximately half would be more accurate
- Table 1 indicates all but one study was located in secondary school - this is worth commenting on - with only one that included university students as well - which is a weird sample to use - I would expect comment on this in the results - is this a gap in primary school that we should be trying to fill?
- The discussion needs extensive work - currently it mainly repeats the results rather than commenting on their implications for policy, practice and future research - the extent to which QE studies overestimate the results indicates the importance of RCTs and needs to be one of the most prominent findings. Having stated that they had the "best" (? most promising results) undermines this important point. The limitations section is better but
Minor comments
The term QE is introduced without definition on page 3
There are several inconsistencies in the text - for example the inclusion criteria is studies performed from 2013 yet the searches were run from Aug 2012 - a study cannot be published before it is performed - additionally in the methods the shorted intervention is 45 minutes but in the discussion it is 50 minutes
Page 9 - line 218 - sentence references Nobre - starts with "other reviews" and yet only one paper is referenced - the sentence itself is unclear
Reviewer 3 Report
First of all, I would like to thank you for the opportunity to review this work.
The authors have done a good job in the Introduction and have provided relevant data to understand the importance of the topic they are trying to address. The objectives are also clear.
As for the Methodology, this paper is a systematic review with meta-analysis, but I have the impression that the authors have left out several relevant papers on this topic. This can probably be explained by the fact that they have not used databases that may contain studies of great importance such as Web of Sciences, among others.
The conclusions drawn are too categorical considering that few documents were used for the meta-analysis. There is much more previous literature on this topic of study and therefore, to be able to obtain adequate results and conclusions under the premise of the chosen documents is too risky.
I believe that the authors have done a good job in terms of conceptualising the document, but they are dealing with a topic that is of great interest and therefore has a lot of literature that has not been taken into account.
Reviewer 4 Report
It is a very important topic that the authors chose and i think it is very important for everyone to educate young school going kids regarding mental health and make them aware that there is help out there and it is very important to educate young people that it is not wrong for them to feel depressed, anxious and it's okay to seek help when they feel down and they should not feel embarassed to seek help. I think it is also important to educate families along with kids.
The Authors did a good job in following PRISMA guidelines and they clearley mentioned the limitaions of such a review.